# Financial hardship among patients suffering from neglected tropical diseases: A systematic review and meta-analysis of global literature

**Chanthawat Patikorn**[1,2], **Jeong-Yeon Cho**[1,3], **Joshua Higashi**[4], **Xiao Xian Huang**[5], **Nathorn Chaiyakunapruk**[1,6]*

**1** Department of Pharmacotherapy, College of Pharmacy, University of Utah, Salt Lake City, Utah, United States of America, **2** Department of Social and Administrative Pharmacy, Faculty of Pharmaceutical Sciences, Chulalongkorn University, Bangkok, Thailand, **3** School of Pharmacy, Sungkyunkwan University, Suwon, South Korea, **4** Corvaxan Foundation, Villanova, Pennsylvania, United States of America, **5** Department of Global Programme for Neglected Tropical Diseases, World Health Organization, Geneva, Switzerland, **6** IDEAS Center, Veterans Affairs Salt Lake City Healthcare System, Salt Lake City, Utah, United States of America

* Nathorn.Chaiyakunapruk@utah.edu

**Data Availability Statement:** The authors confirm that all data underlying the findings are fully available without restriction. All relevant data are

## Abstract

### Introduction

Neglected tropical diseases (NTDs) mainly affect underprivileged populations, potentially resulting in catastrophic health spending (CHS) and impoverishment from out-of-pocket (OOP) costs. This systematic review aimed to summarize the financial hardship caused by NTDs.

### Methods

We searched PubMed, EMBASE, EconLit, OpenGrey, and EBSCO Open Dissertations, for articles reporting financial hardship caused by NTDs from database inception to January 1, 2023. We summarized the study findings and methodological characteristics. Meta-analyses were performed to pool the prevalence of CHS. Heterogeneity was evaluated using the $I^2$ statistic.

### Results

Ten out of 1,768 studies were included, assessing CHS (n = 10) and impoverishment (n = 1) among 2,761 patients with six NTDs (Buruli ulcer, chikungunya, dengue, visceral leishmaniasis, leprosy, and lymphatic filariasis). CHS was defined differently across studies. Prevalence of CHS due to OOP costs was relatively low among patients with leprosy (0.0–11.0%), dengue (12.5%), and lymphatic filariasis (0.0–23.0%), and relatively high among patients with Buruli ulcers (45.6%). Prevalence of CHS varied widely among patients with chikungunya (11.9–99.3%) and visceral leishmaniasis (24.6–91.8%). Meta-analysis showed that the pooled prevalence of CHS due to OOP costs of visceral leishmaniasis was 73% (95% CI; 65–80%, n = 2, $I^2$ = 0.00%). Costs of visceral leishmaniasis impoverished 20–26% of the 61 households investigated, depending on the costs captured. The reported

within the paper and its Supporting Information files.

**Funding:** This study is funded by the Department of Control of Neglected Tropical Diseases, World Health Organization, Geneva, Switzerland. XXH, as an employee of the World Health Organization, contributed to this study in terms of study design, data interpretation, and report writing.

**Competing interests:** I have read the journal's policy and the authors of this manuscript have the following competing interests:XXH works for the World Health Organization. The author alone is responsible for the views expressed in this publication and does not necessarily represent the decisions, policies, or views of the World Health Organization.

costs did not capture the financial burden hidden by the abandonment of seeking healthcare.

## Conclusion

NTDs lead to a substantial number of households facing financial hardship. However, financial hardship caused by NTDs was not comprehensively evaluated in the literature. To develop evidence-informed strategies to minimize the financial hardship caused by NTDs, studies should evaluate the factors contributing to financial hardship across household characteristics, disease stages, and treatment-seeking behaviors.

## Author summary

Neglected tropical diseases (NTDs) mainly affect underprivileged populations, potentially resulting in catastrophic health spending (CHS) and impoverishment from out-of-pocket (OOP) costs. This systematic review aimed to summarize the financial hardship caused by NTDs. We found that NTDs lead to a substantial number of households facing financial hardship. CHS risk due to direct OOP costs was relatively low among patients with leprosy (0.0–11.0%), dengue (12.5%), and lymphatic filariasis (0.0–23.0%), and relatively high among patients with Buruli ulcers (45.6%). CHS risk varied widely among patients with chikungunya (11.9–99.3%) and visceral leishmaniasis (24.6–91.8%). Costs of visceral leishmaniasis impoverished 20–26% of 61 households, depending on the costs captured. Nevertheless, financial hardship caused by NTDs was not comprehensively evaluated in the literature. Therefore, to develop evidence-informed strategies to minimize the financial hardship caused by NTDs, studies should evaluate the factors contributing to financial hardship across household characteristics, disease stages, and treatment-seeking behaviors.

## Introduction

In 2021, the World Health Organization (WHO) reported that 1.65 billion people required treatment and care for neglected tropical diseases (NTDs) as they faced humanistic, social, and economic burdens incurred by the diseases. NTDs are a diverse group of diseases that mainly affect underprivileged communities in tropical and subtropical areas [1]. NTDs predominantly affect disadvantaged populations in low- and middle-income countries (LMICs) due to the lack of timely access to affordable care. It has been reported that every low-income country is affected by at least five NTDs [2]. Even worse, impoverishment serves as a structural determinant. At the same time, it is a consequence of NTDs due to the direct and indirect costs incurred [3]. Therefore, the WHO has advocated in their recent NTDs 2021–2023 roadmap that NTDs must be overcome to attain Sustainable Development Goals (SDGs) and ensure Universal Health Coverage (UHC). The NTDs 2021–2030 roadmap targets that 90% of the population at risk are protected against catastrophic out-of-pocket (OOP) health spending caused by NTDs [1].

Financial hardship is usually quantified as catastrophic health spending (CHS) (as known as catastrophic health expenditure) and impoverishment. CHS is the proportion of households with OOP costs incurred by a specific disease that exceed a specific threshold of the total

household income or expenditure (budget share approach) or non-subsistent household expenditure (capacity-to-pay approach). Impoverishment is when the OOP costs push households below the poverty line [4–6]. CHS and impoverishment are well-established indicators for the financial risk protection of the healthcare system, which was an essential dimension of the UHC as indicated under the SDG 3.8.2 indicators [1,7].

Financial hardship poses a greater challenge for individuals affected by NTDs, as they frequently reside in poverty before the onset of the disease. To evaluate the long-term economic risk imposed by health spending on NTDs, it is important to understand the coping strategies of this population. Literature has shown that coping strategies, such as seeking financial assistance through loans or selling their assets, could push households into or further into poverty if it impacts their productivity [8]. Thus, providing coverage to these groups effectively strengthens the financial risk protection of the health system [7]. Since some types of NTD are closely related to financial hardship, improving their financial protection may help attain UHC, especially for LMICs [9].

Financial protection is an essential indicator for NTDs and UHC; however, there was limited research on the financial hardship of NTDs. Although many studies addressed the question of the economic burden of NTDs, there is no systematic review and meta-analysis summarizing the financial hardship faced by the population affected by NTDs. Therefore, to fill this knowledge gap and build a baseline for the NTDs roadmap's financial risk protection indicator, this study aimed to summarize the prevalence and magnitude of financial hardship among patients suffering from NTDs. Additionally, we assessed the methodologies of quantifying CHS and impoverishment incurred by NTDs.

## Methods

### Scope of the review

The protocol of this systematic review was registered with PROSPERO (CRD42023385627) [10]. This study was reported following the 2020 Preferred Reporting Items for Systematic Reviews and Meta-analyses (PRISMA) reporting guideline (S1 PRISMA Checklist) [11]. Differences from the original review protocol are described with rationale (S1 Table).

This systematic literature review focused on 20 diseases selected as NTDs by WHO: Buruli ulcer, Chagas disease, dengue and chikungunya, dracunculiasis (Guinea-worm disease), echinococcosis, foodborne trematodiases, human African trypanosomiasis (sleeping sickness), leishmaniasis, leprosy (Hansen's disease), lymphatic filariasis, mycetoma, chromoblastomycosis and other deep mycoses, onchocerciasis (river blindness), rabies, scabies and other ectoparasitoses, schistosomiasis, soil-transmitted helminthiases, snakebite envenoming, taeniasis/cysticercosis, trachoma, and yaws and other endemic treponematoses [12].

Outcomes of interest of this systematic review were the prevalence and magnitude of victims who faced financial hardship caused by NTDs, including CHS, impoverishment, and coping strategies.

### Search strategy and selection process

We searched three bibliographic databases, PubMed, EMBASE, and EconLit, to identify articles reporting financial hardship among patients suffering from NTDs from any country indexed from database inception to January 1, 2023. We also searched for grey literature in two databases, OpenGrey and EBSCO Open Dissertations. The search terms used were (*Disease name and its synonyms*) AND (catastroph* OR impoverish* OR coping OR economic consequence* OR out-of-pocket OR "out of pocket" OR ((household OR family OR patient AND (cost* OR spending OR expen*)))), that was adapted to match the search techniques of

each database. A full search strategy is shown in S2 Table. There was no language restriction applied in this systematic review. A supplemental search was performed by tracking citation and snowballing the eligible articles' reference list.

Two reviewers (CP and JYC) independently performed the study selection. They screened the titles and abstracts of identified articles from database searches for relevance. Potentially relevant articles were sought for full-text articles. We requested the authors for full-text articles or reports of highly relevant articles without full-text articles, such as conference abstracts. The retrieved full-text articles were selected based on the eligibility criteria. Discrepancies arising during study selection were resolved by discussion with the third reviewer (NC).

### Eligibility criteria

We included empirical studies reporting CHS, impoverishment, or coping strategies incurred by NTDs using primary data collection.

### Data extraction

We developed a data extraction sheet by performing a pilot test of extracting five randomly selected articles and refining it until finalization. Two reviewers (CP and JYC) independently performed data extraction. Another reviewer (JH) checked the extracted data for correctness. Any discrepancies were resolved by discussion among reviewers.

Study findings and methodological characteristics extracted from the eligible articles are as follows: first author, publication year, NTDs, study setting, study design, sample characteristics, sample size, data collection period, data collection methods, time horizon, a perspective of the analysis, discount rate, costing year, reported currency, cost units, the definition of CHS and impoverishment, prevalence and magnitude of CHS and impoverishment incurred, economic consequences and coping strategies of financial hardship. Corresponding authors of the eligible articles were contacted to request individual patient-level data. However, we received no response.

The financial risk protection metric is intended to capture only the OOP costs for medical services (e.g., treatment and diagnosis costs). However, some studies considered certain types of direct non-medical costs (e.g., transportation, food, and accommodation costs) and indirect costs (e.g., productivity and income losses) when quantifying financial hardship. Some studies also included informal care costs, such as traditional medicine, as OOP costs [6]. Thus, our systematic review categorized costs extracted from the eligible studies as direct costs (OOP costs) and indirect costs. Direct costs were further categorized as direct medical costs and direct non-medical costs. The combination of direct costs and indirect costs was categorized as total costs.

### Quality assessment

Two reviewers independently assessed the eligible articles' quality (CP and JYC). Any discrepancies were resolved by consensus among the reviewers. To the best of our knowledge, there is no risk-of-bias assessment tool for economic burden studies. Hence, we assessed the quality of the eligible articles using the cost-of-illness evaluation checklist by Larg and Moss [13].

### Data synthesis

A narrative synthesis was performed to summarize study findings, methodological characteristics, and the quality of the eligible studies. The identified countries were categorized based on the World Bank's income levels and regions [14].

## Statistical analysis

We performed meta-analyses to calculate the pooled prevalence of households experiencing financial hardship. However, this was possible only for studies that quantified financial hardship using the same measurement definition for a particular NTD. For example, we performed a meta-analysis to calculate the pooled prevalence of households experiencing CHS due to visceral leishmaniasis based on two studies that defined CHS as direct costs exceeding 10% of annual household income [8,15]. The remaining studies were not meta-analyzed due to the differences in the definitions of CHS. We estimated the pooled prevalence of CHS and 95% confidence intervals (CI) using a random-effects model under the DerSimonian and Laird approach [16]. Effect sizes were computed using each study's Freeman–Tukey double-arcsine-transformed proportion. This variance-stabilizing transformation is particularly preferable when the proportions are close to 0 or 1 [17]. $p < .05$ was considered statistically significant in 2-sided tests.

Heterogeneity was evaluated by observing the forest plots and using the $I^2$ statistic that estimated the proportion of variability in a meta-analysis that is explained by differences between the included trials rather than by sampling error. Subgroup analyses were performed to explore possible causes of heterogeneity among study results. Publication bias was assessed using the funnel plot asymmetry test and the Egger regression asymmetry test [18]. Statistical analyses were conducted using Stata version 18.0 (Stata Corporation).

## Patient and public involvement

Patients or the public were not involved in the design, or conduct, or reporting, or dissemination plans of our research.

# Results

## Overall characteristics of the included studies

A total of 1,768 articles were identified from the search, of which 10 studies were included (Fig 1) [8,15,19–26]. A list of excluded studies with reasons is presented in S3 Table. These studies quantified financial hardship among 2,761 patients in five LMICs (India, Nepal, Nigeria, Sudan, and Vietnam) who had been diagnosed with six out of the WHO's 20 NTDs, including Buruli ulcer [20], chikungunya [21,26], dengue [22], visceral leishmaniasis [8,15,25], leprosy [19,23], and lymphatic filariasis [24]. Table 1 provides a summary of the study characteristics. We found no major concern in the quality of the included studies (S4 Table)

Financial hardship caused by NTDs was quantified as CHS (10 studies) [8,15,19–26], and impoverishment (1 study) [8]. All studies were conducted in LMICs with a focus on South Asia (7 studies) [8,19,21,23–26], Sub-Saharan Africa (2 studies) [15,20], East Asia & Pacific (1 study) [22]. Patients were mostly identified using a hospital-based approach (7 studies) [8,15,19,20,22,23,25], with active case-finding intervention implemented in two of those studies [20,23]. Five studies reported that patients sought informal healthcare, such as traditional healers, ayurveda, and homeopathy [19–21,25,26].

Costs captured in the financial hardship were direct medical costs (10 studies, 100%) [8,15,19–26], direct non-medical costs (9 studies, 90%) [8,15,19–21,23–26], and indirect costs (7 studies, 70%) [8,15,19,21,23,25,26], as summarized in Table 2. These costs were captured with a different timeframe, including during a disease episode [8,15,20,21,25,26], during hospitalization in an intensive care unit [22], monthly costs with a maximum recall period of 3 years [19], per one outpatient visit in the last 6 months [23], and per one hospitalization

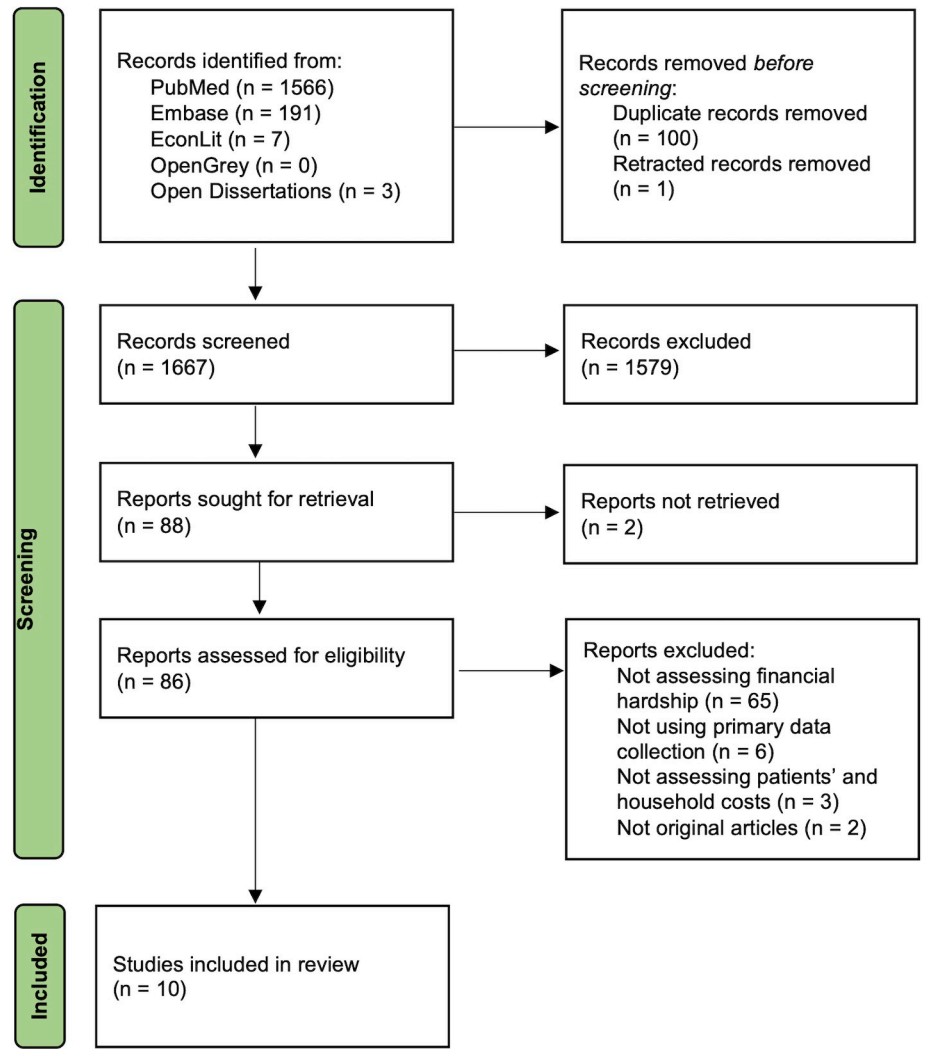

**Fig 1. Study selection flow.**

episode in the last year and per one outpatient visit in the last 15 days [24]. Abandonment of healthcare seeking due to financial burden was not reflected in the reported costs as the included studies captured only patients who sought healthcare.

The health insurance systems or special programs covered some of the costs. The costs for diagnosis and treatment of visceral leishmaniasis were provided free of charge to patients under the publicly financed health insurance system in Nepal [8,25] and Sudan [15]. In Nigeria, international development partners funded a special program that provided free diagnosis and treatment of Buruli ulcers, as well as accommodation, school funding, and basic allowance [20]. Additionally, the Indian government had a special program that provides financial assistance to families of patients affected by leprosy [19]. However, patients in India had to pay high OOP costs for medical services for leprosy [19,23], chikungunya [21,26], and lymphatic filariasis [24]. Similarly, patients in Vietnam also paid high OOP costs for the medical treatment of dengue [22]. For more details, refer to Table 3.

**Table 1. Characteristics of studies assessing financial hardship.**

| First author, Year of publication | NTDs | Region | Country | Income economy | Study population (Sample size) | Case identification approach | Treatment seeking behavior |
|---|---|---|---|---|---|---|---|
| Chukwu, 2017 [20] | Buruli ulcer | Sub-Saharan Africa | Nigeria | Lower middle | Laboratory-confirmed patients with buruli ulcer in four States (Cross River, Anambra, Imo, and Ogun) in Southern Nigeria during July to September 2015 (n = 92) | Hospital-based with active case-finding intervention | Before diagnosis<br>- 82% Patent medicine dealer/vendor<br>- 72% Traditional medicine practitioner<br>- 34% Prayer house/faith-healing<br>- 28% Primary health center<br>- 27% Public secondary-care hospital<br>- 19% Private hospital<br>- 11% Mission hospital |
| Gopalan, 2009 [21] | Chikungunya | South Asia | India | Lower middle | Bread winners of the household who had developed sudden onset fever and bodyache during chikungunya outbreak and who had already completed the treatment in Kural village in Nayagarh district of Orissa state India during May to July 2007 (n = 150) | Community-based | Any visits<br>49% Private hospital only<br>31% Public and private hospitals<br>20% Public hospital only<br>Majority of private providers were allopathy, ayurveda, homeopathy, traditional healers, and informal service providers (quacks) |
| Vijayakumar, 2013 [26] | Chikungunya | South Asia | India | Lower middle | Patients who had suffered from chikungunya during chikungunya outbreak in 2007 in five districts (Kollam, Alappuzha, Kottayam, Pathanamthitta, Iddukki) in Kerala India (n = 1822) | Community-based | Any visits<br>92% Modern medicines only<br>46% Government facilities<br>44% Private facilities<br>4% Ayurveda or Homeopathy<br>4% Combination |
| McBride, 2019 [22] | Dengue | East Asia & Pacific | Vietnam | Lower middle | Patients with dengue shock who were treated in intensive care unit at the Hospital for Tropical Diseases, a tertiary referral hospital for infectious diseases in Ho Chi Minh City, Vietnam during November 2014 to January 2016 (n = 88) | Hospital-based | Not reported |
| Adhikari, 2009 [8] | Visceral leishmaniasis | South Asia | Nepal | Lower middle | Laboratory-confirmed patients with visceral leishmaniasis in Siraha and Saptari districts, Nepal during February 2004 (n = 61) | Hospital-based | Not reported |
| Meheus, 2013 [15] | Visceral leishmaniasis | Sub-Saharan Africa | Sudan | Low | Laboratory-confirmed patients with visceral leishmaniasis hospitalized in three public hospitals in Gedaref State, Sudan during December 2010 to May 2011 (n = 75) | Hospital-based | First visit<br>43% Public provider at village health worker<br>25% Public hospital<br>20% Public health center<br>9% Private general practitioner<br>3% Chemist |
| Uranw, 2013 [25] | Visceral leishmaniasis | South Asia | Nepal | Lower middle | Laboratory-confirmed patients with visceral leismaniasis five districts (Siraha, Saptari, Sunsari, Morang and Jhapa) in south-eastern Nepal during August to September 2010 (n = 168) | Hospital-based | First visit<br>55% Public provider<br>20% Private provider<br>15% Traditional healer<br>10% Chemist or pharmacy |
| Chandler, 2015 [19] | Leprosy | South Asia | India | Lower middle | Patients with lepromatous and borderline lepromatous leprosy with ENL (n = 53) or without (n = 38) who attended a leprosy hospital in Purulia district of West Benga, India during June to July 2013 (N = 91) | Hospital-based | ENL<br>64% Private hospitals<br>43% Traditional healers<br>No ENL<br>47% Private hospitals<br>29% Traditional healers |

*(Continued)*

**Table 1.** (Continued)

| First author, Year of publication | NTDs | Region | Country | Income economy | Study population (Sample size) | Case identification approach | Treatment seeking behavior |
|---|---|---|---|---|---|---|---|
| Tiwari, 2018 [23] | Leprosy | South Asia | India | Lower middle | Patients with leprosy in two public health settings (the Union Territory of Dadra and Nagar Haveli [n = 103] and the Umbergaon block of Valsad, Gujrat [n = 37]) during May to October 2016 (N = 140) | Hospital-based with active case-finding intervention | Last 3 visits in 6 months<br>80% Government only<br>14% Private only<br>6% Both |
| Tripathy, 2020 [24] | Lymphatic filariasis | South Asia | India | Lower middle | Hospitalization episodes of lymphatic filariasis (n = 38) and episodes of outpatient care for lymphatic filariasis (n = 36) in India which were identified from the National Sample Survey Organization in 2014 (N = 74) | Community-based nationwide survey | Inpatient visit<br>50% Private<br>47% Public<br>Outpatient visit<br>72% Private<br>22% Public |

Note: Total costs comprise direct and indirect costs. Abbreviations: DC–direct costs; ENL—erythema nodosum leprosum; NTDs–neglected tropical diseases; TC–total costs.

### Financial hardship among patients suffering from NTDs

**Catastrophic health spending.** CHS was variedly defined across studies in terms of types of costs (medical costs, medical and transportation costs, direct costs, indirect costs, or total costs), thresholds (5%, 10%, 15%, 25%, 30%, 40%, or 100%), timeframe (monthly, quarterly, or annual), household resources (income, consumption expenditure, national average annual household expenditure, or international poverty line) and perspective (household or individual). All studies used the budget share approach to quantify CHS. The most commonly used definitions of CHS caused by NTDs were direct costs of a disease episode exceeding 10% of annual household income (3 studies) [8,15,20] and total costs of a disease episode exceeding 10% of annual household income (3 studies) [8,15,25]. CHS that included only the direct medical costs was reported in two studies [8,22].

We summarized the prevalence of households experiencing CHS and the magnitude of CHS, determined as the percentage of the costs of NTDs as a share of income, in Table 4. The prevalence and magnitude of CHS varied depending on the definitions of CHS, disease duration (episodic or chronic), and thresholds used ($\leq$10% or >10%). Overall, the direct costs of NTDs resulted in a wide range of households experiencing CHS. CHS was generally low among patients with leprosy (0.0–11.0%) [19,23], dengue (12.5%) [22], and lymphatic filariasis (0.0–23.0%) [24], and relatively high among patients with Buruli ulcers (45.6%) [20]. CHS varied widely among patients with chikungunya (11.9–99.3%) [21,26] and visceral leishmaniasis (24.6–91.8%) [8,15,25].

Meta-analyses were performed to pool the prevalence of CHS in studies reporting CHS using the same measurement definition in a particular CHS. This was only possible for visceral leishmaniasis, in which CHS was quantified as direct costs of a disease episode exceeding 10% of annual household income in two studies [8,15], and total costs exceeding 10% of annual household income in three studies [8,15,25].

The pooled prevalence of CHS, defined as direct costs exceeding 10% of annual household income, was 73% (95% CI; 65–80%, n = 2, $I^2$ = 0.00%), as shown in Fig 2A. Egger's test (P = 0.80) indicated no evidence of small-study effects. Visual inspection of the funnel plot indicated no evidence of publication bias (S1A Fig).

**Table 2. Financial hardship among patients suffering from neglected tropical diseases.**

| First author, Year of publication | NTDs | Timeframe of costs captured | Share of costs out of household income, % | % Households experiencing catastrophic health spending | % Households experiencing impoverishment | % Coping strategies of households |
|---|---|---|---|---|---|---|
| Chukwu, 2017 [20] | Buruli ulcer | Illness onset to treatment completion | 13%: Mean DC out of median annual household income | 50%: DC > 10% annual household income | Not reported | Not reported |
| Gopalan, 2009 [21] | Chikungunya | Illness onset to treatment completion | 37%: Median DC out of median monthly household income | 99%: DC > 10% monthly household income | Not reported | Not reported |
| Vijayakumar, 2013 [26] | Chikungunya | Illness onset to treatment completion | 9% Median DC out of median monthly individual income | - 25%: DC > monthly individual income <br> - 12%: DC > monthly international poverty line | Not reported | Not reported |
| McBride, 2019 [22] | Dengue | During hospitalization in intensive care unit | Not applicable: Household income not reported | 13%: Medical costs per hospitalization > 10% national average annual household expenditure | Not reported | Not reported |
| Adhikari, 2009 [8] | Visceral leishmaniasis | Illness onset to treatment completion | - 17%: Mean DC out of mean annual household income <br> - 27%: Mean IC out of mean annual household income <br> - 44%: Mean TC out of mean annual household income | Threshold at 5% <br> - 75%: Medical costs > 5% annual household income <br> - 82%: Medical and transportation costs > 5% annual household income <br> - 92%: DC > 5% annual household income <br> - 93%: TC > 5% annual household income <br> Threshold at 10% <br> - 49%: Medical costs > 10% annual household income <br> - 61%: Medical and transportation costs > 10% annual household income <br> - 70%: DC > 10% annual household income <br> - 85%: TC > 10% annual household income <br> Threshold at 15% <br> - 31%: Medical costs > 15% annual household income <br> - 41%: Medical and transportation costs > 15% annual household income <br> - 54%: DC > 15% annual household income <br> - 69%: TC > 15% annual household income <br> Threshold at 25% <br> - 10%: Medical costs > 25% annual household income <br> - 15%: Medical and transportation costs > 25% annual household income <br> - 25%: DC > 25% annual household income <br> - 52%: TC > 25% annual household income | - 20%: Annual household income after medical costs fell below poverty line <br> - 21%: Annual household income after medical and transportation costs fell below poverty line <br> - 26%: Annual household income after DC fell below poverty line | 80%: Took a loan |
| Meheus, 2013 [15] | Visceral leishmaniasis | Illness onset to treatment completion | 23%: Median TC out of median annual household income | - 75%: DC > 10% annual household income <br> - 83%: TC > 10% annual household income | Not reported | Not reported |

*(Continued)*

**Table 2.** (Continued)

| First author, Year of publication | NTDs | Timeframe of costs captured | Share of costs out of household income, % | % Households experiencing catastrophic health spending | % Households experiencing impoverishment | % Coping strategies of households |
|---|---|---|---|---|---|---|
| Uranw, 2013 [25] | Visceral leishmaniasis | Illness onset to treatment completion | 11% Median TC out of median annual household income | 51%: TC > 10% annual household income | Not reported | - 71%: Used savings<br>- 56%: Took a loan<br>- 17%: Sold livestocks<br>- 42%: Used any two strategies<br>- 2%: Used all three strategies |
| Chandler, 2015 [19] | Leprosy | Monthly costs with maximum recall period of 3 years | ENL<br>- 8%: Median monthly DC out of median monthly household income<br>- 18%: Median monthly IC out of median monthly household income<br>- 28%: Median monthly TC out of median monthly household income<br>No ENL<br>- 4% Median monthly DC out of median monthly household income<br>- 1% Median monthly IC out of median monthly household income<br>- 5% Median monthly TC out of median monthly household income | ENL<br>- 11%: Monthly DC > 40% monthly household income<br>- 38%: Monthly TC > 40% monthly household income<br>No ENL<br>- 0%: Monthly DC > 40% monthly household income<br>- 3%: Monthly TC > 40% monthly household income | Not reported | ENL<br>- 100%: Used cash savings<br>- 70%: Sold assets, borrowed money, or being gifted money<br>- 42%: Took a loan<br>- 32%: Sold assets<br>No ENL<br>- 100%: Used cash savings<br>- 55%: Sold assets, borrowed money, or being gifted money<br>- 32%: Took a loan<br>- 17%: Sold assets |
| Tiwari, 2018 [23] | Leprosy | Per outpatient visit in the last 6 months | 4%: Average of % DC per outpatient visit out of quarterly individual income | 6%: DC per outpatient visit > 10% quarterly individual income | Not reported | Not reported |
| Tripathy, 2020 [24] | Lymphatic filariasis | - Per hospitalized episode in the last year<br>- Per outpatient visit in the last 15 days | Inpatient visit<br>14%: Median DC out of median annual household consumption expenditures<br>Outpatient visit<br>0.5%: Median DC out of median annual household consumption expenditures | Inpatient visit<br>23%: DC per hospitalization > 30% annual household consumption expenditures<br>Outpatient visit<br>0%: DC per outpatient visit > 30% annual household consumption expenditures | Not reported | Inpatient visit<br>23%: Borrowed or sold assets<br>Outpatient visit<br>0%: Borrowed or sold assets |

Note: Total costs comprise direct and indirect costs. Abbreviations: DC–direct out-of-pocket costs; ENL—erythema nodosum leprosum; IC–indirect costs; NTDs–neglected tropical diseases; TC–total costs.

**Table 3. Details of costs incurred from neglected tropical diseases.**

| First author, Year of publication | NTDs | Costs covered by national health insurance | Components of direct medical costs | Components of direct non-medical costs | Components of Indirect costs | Costs out of total costs, % | | |
|---|---|---|---|---|---|---|---|---|
| | | | | | | Direct medical costs | Direct non-medical costs | Indirect costs |
| Chukwu, 2017 [20] | Buruli ulcer | - Free diagnosis and treatment of buruli ulcer<br>- Provide accommodation, school funding, and basic allowance | - Medication<br>- Laboratory test<br>- Hospitalization<br>- Informal care<br>- Others (not specified) | - Transportation<br>- Food<br>- Others (not specified) | Not included | 98% | 2% | Not included |
| Gopalan, 2009 [21] | Chikungunya | Medical treatment costs are highly paid out of pocket | - Treatment<br>- Diagnosis<br>- Consultation<br>- Drug<br>- Hospitalization | - Transportation<br>- Stay<br>- Food<br>- Escort | - Lost workdays of the patients<br>- Lost workhours of the patients<br>- Income losses of the patients | 39% | 13% | 47% |
| Vijayakumar, 2013 [26] | Chikungunya | Medical treatment costs are highly paid out of pocket | - Doctor fees<br>- Medicine<br>- Investigation<br>- Others (not specified) | - Transportation<br>- Food | - Lost workdays of the patients and their caretakers<br>- Income losses of household | 27% | 8% | 65% |
| McBride, 2019 [22] | Dengue | Medical treatment costs are highly paid out of pocket | Hospital bill | Not included | Not included | 100% | Not included | Not included |
| Adhikari, 2009 [8] | Visceral leishmaniasis | Free diagnosis and treatment of visceral leishmaniasis | Hospital-based medical care | - Travel<br>- Food<br>- Others (e.g. small offerings to hospital staff at the time of discharge, payments to middlemen for hospital access) | - Lost workdays of household<br>- Income losses of household | 26% | 13% | 61% |
| Meheus, 2013 [15] | Visceral leishmaniasis | Free diagnosis and treatment of visceral leishmaniasis | - Drug<br>- Registration<br>- Laboratory test<br>- Medical supply | - Food<br>- Transportation | - Lost workdays of household<br>- Income losses of household | 26% | 60% | 14% |
| Uranw, 2013 [25] | Visceral leishmaniasis | Free diagnosis and treatment of visceral leishmaniasis | - Consultation<br>- Medicine<br>- Laboratory test | - Transportation<br>- Food<br>- Daily expenditures for the patient and accompanying family members | - Lost workdays of household<br>- Income losses of household | 24% | 23% | 53% |
| Chandler, 2015 [19] | Leprosy | Financial assistance for the families of patients affected by leprosy | - Consultation<br>- Hospital admission<br>- Investigation<br>- Medicines<br>- Other treatments | - Transportation<br>- Additional food<br>- Other non-medical goods or services | - Lost workdays of household<br>- Income losses of household | ENL 24%<br>No ENL 44% | ENL 11%<br>No ENL 35% | ENL 65%<br>No ENL 21% |
| Tiwari, 2018 [23] | Leprosy | Medical treatment costs are highly paid out of pocket | - Consultation<br>- Investigation<br>- Medicine<br>- supply | - Transportation<br>- Food | - Lost workdays of household<br>- Income losses of household | 39% | 6% | 55% |
| Tripathy, 2020 [24] | Lymphatic filariasis | Medical treatment costs are highly paid out of pocket | - Drug<br>- Diagnosis test<br>- Doctor fees<br>- Other medical expenses | - Transportation<br>- Food and lodging for the patient and other accompanying persons | Not included | Inpatient visit 87%<br>Outpatient visit 63% | Inpatient visit 13%<br>Outpatient visit 38% | Not included |

Note: Total costs comprise direct and indirect costs. Abbreviations: ENL—erythema nodosum leprosum; NTDs–neglected tropical diseases.

**Table 4. Summary of prevalence and magnitude of catastrophic health spending.**

| | OOP costs (no. of studies) | | Total costs (no. of studies) | |
|---|---|---|---|---|
| | %CHS | % OOP costs/income | %CHS | %Total costs/income |
| **Disease** | | | | |
| Overall | 0.0–99.3% (n = 9) | 0.5–37.2% (n = 9) | 2.60–93.4% (n = 4) | 4.9–44.4% (n = 4) |
| Buruli ulcer | 45.6% (n = 1) | 13.0% (n = 1) | | |
| Chikungunya | 11.9–99.3% (n = 2) | 9.1–37.2% (n = 2) | | |
| Dengue | 12.5% (n = 1) | Not reported (n = 1) | | |
| Visceral leishmaniasis | 24.6–91.8% (n = 2) | 17.5–23.0% (n = 2) | 52.5–93.4% (n = 3) | 11.0–44.4% (n = 3) |
| Leprosy | 0.0–11.0% (n = 2) | 3.7–7.5% (n = 2) | 2.6–37.7% (n = 1) | 4.9–27.9% (n = 1) |
| Lymphatic filariasis | 0.0–23.0% (n = 1) | 0.5–14.0% (n = 1) | | |
| **Disease duration** | | | | |
| Episodic | 11.9–99.3% (n = 6) | 9.1–37.2% (n = 6) | 51.2–85.3% (n = 3) | 11.0–44.4% (n = 3) |
| Chronic | 0.0–23.0% (n = 3) | 0.5–14.0% (n = 3) | 2.6–37.7% (n = 1) | 4.9–27.9% (n = 1) |
| **Threshold used** | | | | |
| Threshold ≤10% | 5.7–91.8% (n = 6) | 4.5–37.2% (n = 6) | 51.2–93.4% (n = 3) | 11.0–44.4% (n = 3) |
| Threshold >10% | 0.0–54.1% (n = 4) | 0.5–17.5% (n = 4) | 2.6–68.9% (n = 2) | 4.9–44.4% (n = 2) |

Abbreviations: CHS–catastrophic health spending; OOP–out-of-pocket.

The pooled prevalence of CHS, defined as total costs exceeding 10% of annual household income, was 74% (95% CI; 49–93%, n = 3, $I^2$ = 94.72%), as shown in S2 Fig. We explored the source of heterogeneity by visual inspection of the forest plot. We found that the source of heterogeneity was the differences in the treatment of visceral leishmaniasis, where sodium stibogluconate was used in two studies [8,15], and miltefosine in one study [25]. Therefore, we performed a subgroup meta-analysis based on different treatments, as shown in Fig 2B. We removed one study [25] from the meta-analysis to investigate the publication bias without the presence of heterogeneity. Egger's test (P = 0.81) indicated no evidence of small-study effects. Visual inspection of the funnel plot indicated no evidence of publication bias (S1B Fig).

**Impoverishment.** Impoverishment was investigated in one study in patients with visceral leishmaniasis, which defined impoverishment as annual household income falling below the poverty line after paying for treatment [8]. Costs of visceral leishmaniasis impoverished 20–26% of the 61 households investigated, depending on the costs captured (20% medical costs, 21% medical and transportation costs, 26% direct costs), as shown in Table 2.

## Coping strategies

Four studies reported coping strategies used by patients to pay the costs of NTDs. These strategies included using savings (71–100% of patients), taking out loans (32–80%), selling livestock or other assets (17–32%), or borrowing money (0–23%), as shown in Table 2. However, these studies did not distinguish between coping strategies used by patients who experienced CHS and those who did not [8,19,24,25].

## Cost drivers and determinants of financial hardship

To understand the cost drivers of financial hardship caused by NTDs, we analyzed the percentage share of types of costs captured in the direct costs. The findings are presented in Fig 3. Direct medical costs were the primary cost driver in nine studies [8,19–21,23–26]. However, one study identified food and transportation costs as the main cost drivers [15].

Determinants of CHS were assessed in one study among patients with Buruli ulcers. The study concluded that neither age, gender, rural/urban location, education, occupation,

### A. Direct costs (medical and non-medical costs) exceeding 10% of annual household income

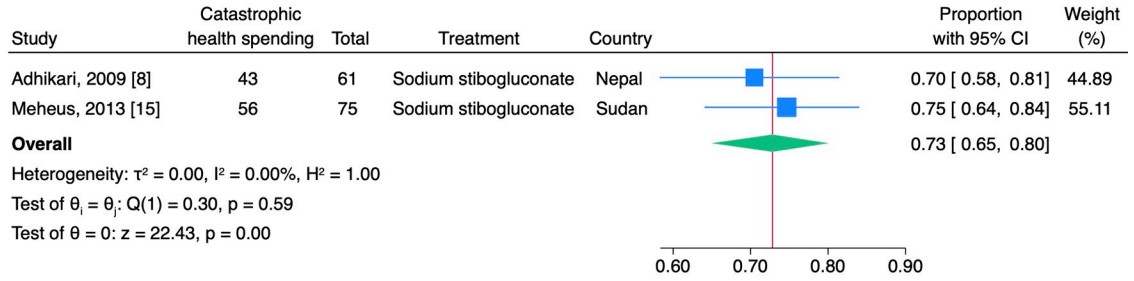

### B. Total costs (direct and indirect costs) exceeding 10% of annual household income

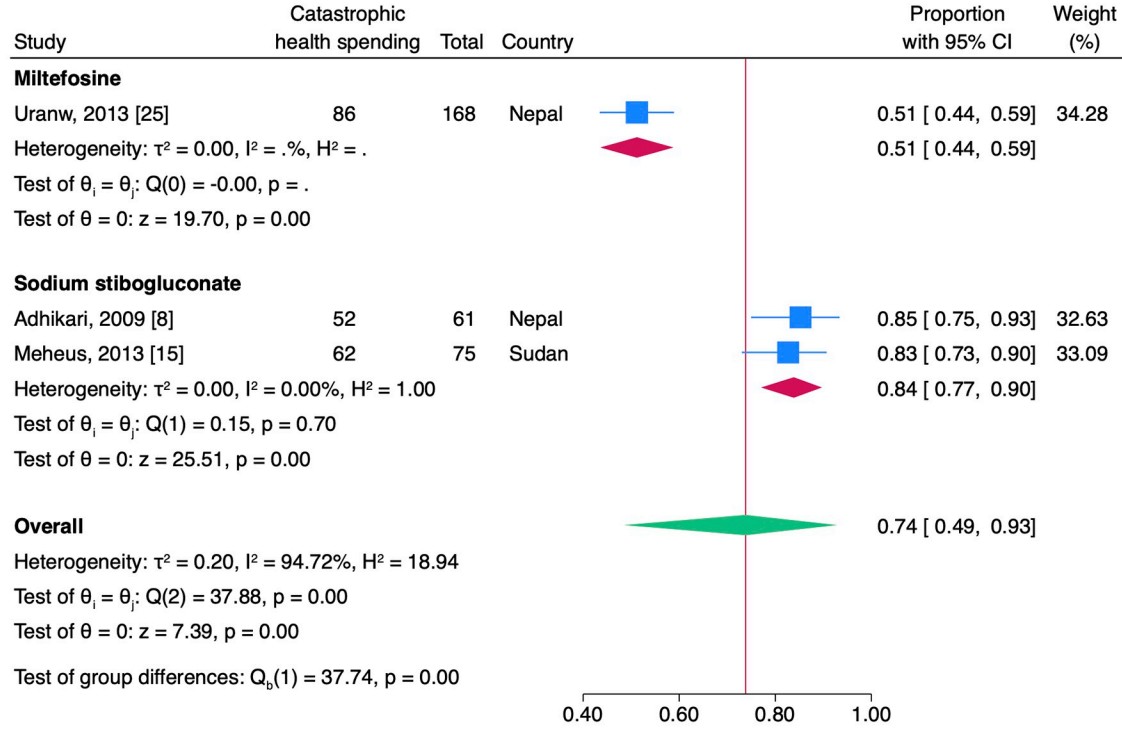

**Fig 2. Meta-analyses of a prevalence of households experiencing catastrophic health spending due to visceral leishmaniasis.**

religion, nor patient income group was a determinant of CHS [20]. There was no study investigating determinants of impoverishment.

## Discussion

NTDs primarily impact populations with limited financial means, yet the literature addressing the financial hardship caused by NTDs is relatively scarce. Our systematic review revealed that

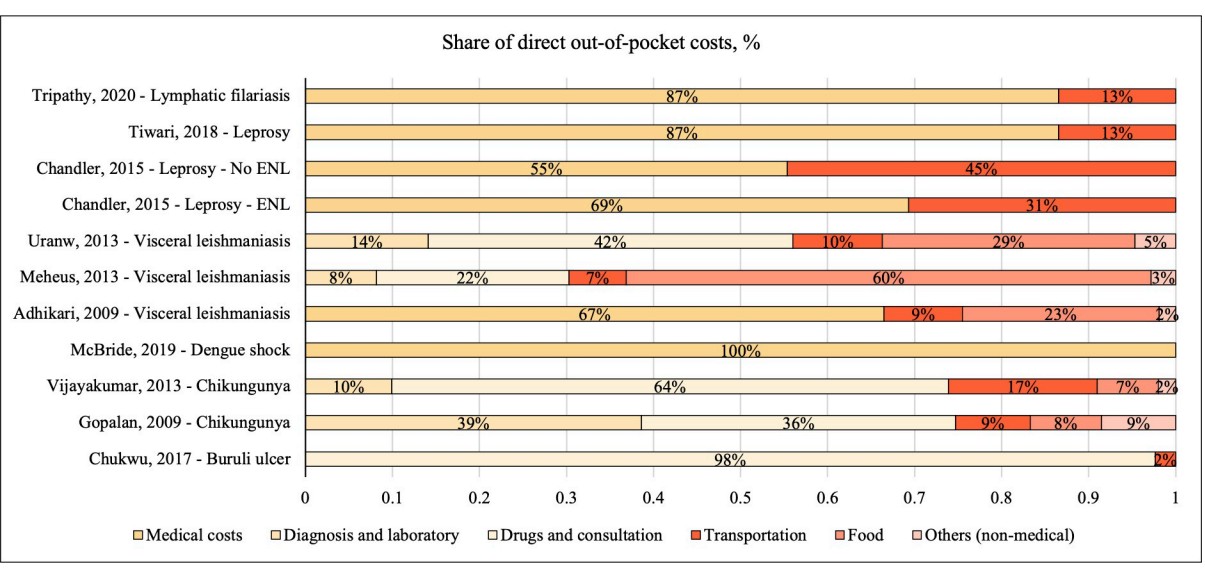

**Fig 3. Cost drivers of out-of-pocket costs.** Abbreviation: ENL–erythema nodosum leprosum. **Tripathy et al, 2020** [24]; **Tiwari et al, 2018** [23]; **Chandler et al, 2015** [19]; **Uranw et al, 2013** [25], **Meheus et al, 2013** [15], **Adhikari et al, 2009** [8], **McBride et al, 2019** [22], **Vijayakumar et al, 2013** [26], **Gopalan et al, 2009** [21], **Chukwu et al, 2017** [20].

there were only ten studies covering six NTDs. We discovered that many households are facing financial hardship as a result of NTDs, despite having access to publicly funded healthcare systems or special NTD programs. The costs related to NTDs resulted in significant financial hardship for these households, mainly due to the high OOP costs associated with medical treatment. Even in situations where drugs used to treat NTDs were provided free of charge, the costs for supportive care, medical procedures, transportation, and food were still high and could have a devastating financial impact on these households. Moreover, these financial hardship indicators might not fully reflect the financial risk of the population affected by NTDs because many live in poverty or even extreme poverty. Victims of NTDs are usually those who are socially disadvantaged. They need to make trade-offs between suffering from the disease and seeking healthcare because not all victims can afford the costs of NTDs, especially OOP costs for medical treatment and transportation, which could lead to the abandonment of healthcare [1–3].

The research findings have shown that merely providing funding for treatments of NTDs is insufficient for protecting those affected by NTDs from financial hardship. Therefore, it is crucial to strengthen the entire healthcare system to effectively address the challenges of NTDs and provide financial protection to the victims. Additionally, it is important to encourage and engage communities to change the behavior of those affected by NTDs so that they seek medical assistance at appropriate healthcare facilities instead of relying on traditional healers or not seeking care at all. Our research also supports the need for an economic framework to guide NTD investments [27]. The ability to prioritize investments, informed partially by economic parameters, may appeal to a broad set of stakeholders and help facilitate the process of building coalitions to achieve the WHO's goal that 90% of the at-risk population is protected against financial hardship caused by NTDs [1].

Although there is no consensus regarding the estimation approach and thresholds in quantifying CHS, it is important to note that these differences can significantly impact the findings and consequently impact the applications and implications of the findings [6,28]. We found that CHS was variedly defined across studies in terms of estimation approach, types of costs, thresholds, timeframe, household resources, and perspective. Our review revealed that 90% of

the included studies captured direct non-medical costs as part of the OOP costs [8,15,19–21,23–26]. Furthermore, Seventy percent of the included studies considered indirect costs in quantifying financial hardship [8,15,19,21,23,25,26]. This approach aligned with an indicator called "catastrophic costs" that has emerged in tuberculosis studies. Catastrophic costs occur when the total healthcare costs, including direct and indirect costs, exceed 20% of the annual household income [28]. This indicator could be a more comprehensive measure of the overall financial burden of NTDs on the household beyond just the OOP costs which will be useful when evaluating and monitoring different healthcare policies and interventions to mitigate financial hardship caused by NTDs.

The findings of this systematic review and meta-analysis should be interpreted under the following limitations. The included studies in our review only focused on patients who sought healthcare, so the financial burden of those who did not seek healthcare was not captured in the reported OOP costs. This means that people who could not afford healthcare may have been excluded from these studies. Moreover, we could not perform meta-analyses of the prevalence of CHS on all identified NTDs due to differences in how CHS was quantified across studies and lack of access to individual patient-level data.

Hence, we highlighted some methodological considerations to guide future studies on financial hardship among households suffering from NTDs to gain a better understanding of the neglected public health issues and to inform the development of strategies of what to address to tackle the financial burden of NTDs. Firstly, methods to quantify financial hardship should be coherent to allow comparability across studies. For instance, CHS and impoverishment should be defined and measured in a relevant manner to the nature of the NTD, including estimation approach, thresholds, types of costs, timeframe, household resources, and perspective. Secondly, subgroup analyses should be conducted to evaluate the determinants of financial hardship across household characteristics (e.g., income, socioeconomic status) or phases of disease (e.g., disease onset, treatment seeking, diagnosis, treatment, post-treatment). Lastly, coping strategies should be assessed among those who did and did not experience financial hardship to understand the economic consequences of financial hardship across subgroups.

## Conclusion

NTDs can be a devastating burden on households, not only in terms of physical and mental health but also financially. NTDs lead to a substantial number of households facing financial hardship. However, financial hardship caused by NTDs was not comprehensively evaluated in the literature. Furthermore, OOP costs represented only a partial picture of the financial hardship the population affected by NTDs faces. To mitigate this financial hardship, it is imperative to conduct thorough research to identify the factors contributing to it. Future research should consider various household characteristics, such as income, education level, and geographic location, as well as the different disease stages, from onset to treatment completion. Future studies should also investigate the hidden financial burden due to the abandonment of healthcare-seeking to capture the economic burden and opportunity costs of those who did not seek healthcare. By carefully examining these factors, researchers and decision-makers can gain insight into the specific challenges faced by households affected by NTDs and develop targeted interventions to alleviate financial hardships. Ultimately, these studies can help inform the development of strategies to reduce the burden of NTDs on households and improve overall health outcomes.

## Supporting information

**S1 PRISMA Checklist. Prisma Checklist.**
(DOCX)

**S1 Table. Differences from original review protocol.**
(DOCX)

**S2 Table. Full search strategy.**
(DOCX)

**S3 Table. Excluded studies with reasons.**
(DOCX)

**S4 Table. Quality assessment using Larg, A., and Moss, J. R. (2011) Cost-of-illness studies: a guide to critical evaluation.**
(DOCX)

**S1 Fig. Assessment of publication bias.**
(TIFF)

**S2 Fig. Forest plot of pooled proportion of catastrophic health spending defined as total costs exceeding 10% of annual household income.**
(TIFF)

## Acknowledgments

The authors alone are responsible for the views expressed in this article and they do not necessarily represent the views, decisions or policies of the institutions with which they are affiliated.

## Author Contributions

**Conceptualization:** Chanthawat Patikorn, Jeong-Yeon Cho, Xiao Xian Huang, Nathorn Chaiyakunapruk.

**Data curation:** Chanthawat Patikorn, Jeong-Yeon Cho.

**Formal analysis:** Chanthawat Patikorn, Jeong-Yeon Cho.

**Funding acquisition:** Chanthawat Patikorn, Nathorn Chaiyakunapruk.

**Investigation:** Chanthawat Patikorn, Jeong-Yeon Cho, Joshua Higashi.

**Methodology:** Chanthawat Patikorn.

**Project administration:** Chanthawat Patikorn, Nathorn Chaiyakunapruk.

**Resources:** Chanthawat Patikorn.

**Software:** Chanthawat Patikorn.

**Validation:** Jeong-Yeon Cho.

**Visualization:** Chanthawat Patikorn.

**Writing – original draft:** Chanthawat Patikorn.

**Writing – review & editing:** Jeong-Yeon Cho, Joshua Higashi, Xiao Xian Huang, Nathorn Chaiyakunapruk.

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
