## [Decision Letter · Decision Letter 0]

24 Jan 2024

Dear Prof. Chaiyakunapruk,

Thank you very much for submitting your manuscript "Financial Catastrophe among Patients Suffering from Neglected Tropical Diseases: A Systematic Review of Global Literature" for consideration at PLOS Neglected Tropical Diseases. As with all papers reviewed by the journal, your manuscript was reviewed by members of the editorial board and by several independent reviewers. The reviewers appreciated the attention to an important topic. Based on the reviews, we are likely to accept this manuscript for publication, providing that you modify the manuscript according to the review recommendations. 

Apologies for the long delay in reaching a decision, I was waiting on a third reviewer but they eventually withdrew their offer to review the manuscript. The two other reviewers however have provided useful feedback - all relatively minor suggested changes that will improve the clarity of the paper.

Sincerely,

Yoel Lubell

Guest Editor

Justin Remais

Section Editor

Apologies for the long delay in reaching a decision, I was waiting on a third reviewer but they eventually withdrew their offer to review the manuscript. The two other reviewers however have provided useful feedback - all relatively minor suggested changes that will improve the clarity of the paper.

Reviewer's Responses to Questions

**Key Review Criteria Required for Acceptance?**

**Methods**

-Are the objectives of the study clearly articulated with a clear testable hypothesis stated?

-Is the study design appropriate to address the stated objectives?

-Is the population clearly described and appropriate for the hypothesis being tested?

-Is the sample size sufficient to ensure adequate power to address the hypothesis being tested?

-Were correct statistical analysis used to support conclusions?

-Are there concerns about ethical or regulatory requirements being met?

Reviewer #1: Yes

Reviewer #2: (No Response)

**Results**

-Does the analysis presented match the analysis plan?

-Are the results clearly and completely presented?

-Are the figures (Tables, Images) of sufficient quality for clarity?

Reviewer #1: Yes

Reviewer #2: (No Response)

**Conclusions**

-Are the conclusions supported by the data presented?

-Are the limitations of analysis clearly described?

-Do the authors discuss how these data can be helpful to advance our understanding of the topic under study?

-Is public health relevance addressed?

Reviewer #1: Yes

Reviewer #2: (No Response)

**Editorial and Data Presentation Modifications?**

Reviewer #1: Major Revision

Reviewer #2: (No Response)

**Summary and General Comments**

Reviewer #1: I read the manuscript with interest and the study poses very important research question. I have few below-mentioned suggestions for the improvement of the manuscript.

1) In abstract, line "Meta-analysis showed that CHE risk due to VL was 73% (95% CI; 65–80%, n = 2, I2 = 0.00%)," I2 may not be clear to readers of the journal. Please specify what does I2 denotes.

2) In abstract, line "Costs of VL impoverished approximately one-fifth of households." Mention N.

3) Introduction needs to be elaborated more to highlight the rationale and importance of the study. The outcomes of the study include CHE, impoverishment, and coping strategies as well. However, in introduction section only CHE has been explained. 

4) The manuscript contains many grammatical errors. Authors are advised to use professional editing tools to rectify all such errors.

5) Line 75-77, "In 2021, the World Health Organization (WHO) reported that 1.65 billion people required mass or individual treatment and care for neglected tropical diseases (NTDs) as they faced human, social, and economic burdens incurred by the diseases" What do the author mean by human burden incurred by the diseases? 

6) Line 77-78, "NTDs are defined as a diverse group of diseases whose impact on impoverished communities in (sub-) tropical areas.(1)" is incomplete and poorly framed.

7) Line 80-82, "For example, as in the study reported before the first road map was published, every low-income country was affected by at least five NTDs.(2)" The line is difficult to comprehend. 

8) Line 183-185, "We performed meta-analyses to calculate the pooled proportions of CHE, which were measured using the same definition, e.g., direct OOP costs exceeded 10% of annual household income." In literature, there is no consensus regarding the threshold at which CHE is calculated. Were there any studies that estimated CHE at any other threshold (say 15% or 25%) of household's income? Were there any studies that estimated CHE using 40% of the household's capacity to pay approach? How were such studies dealt with?

9) Line 230-233. "Health systems provided the diagnosis and treatment of NTDs free of charge(16, 18, 21, 24) or financial assistance to households,(17, 18) while medical treatment costs of NTDs were highly paid OOP in the other health systems,(19, 20, 22, 23, 25) as presented in Table 3." Authors must mention which health systems (e.g., publicly financed) provided free of charge treatment.

10) The discussion section needs to be elaborated to highlight important policy implications and recommendations.

Reviewer #2: This paper describes a systematic review of the Financial Catastrophe among Patients Suffering from Neglected Tropical Diseases. Overall, I found that the paper is clearly written and well conducted study

I have the following comments/suggestions

In my experience, CHE calculations should traditionally use direct costs within the OPP payment component. I have seen papers also include indirect costs within the OPP payment but personally I’m not sure that this is consistent with the foundation of CHE calculations. In the TB literature, they refer to estimates that include indirect costs as catastrophic cost rather than CHE. Assuming I’m not wrong, I think this would be helpful to explore this inconsistency a little more in the discussion. 

There are two main approaches to CHE calculations. Budget share vs capacity to pay. The results could include which one was used and the differences discussed 

I would consider adding Meta-analysis into the title 

I would say that dengue is the NTD and dengue shock is one of its conditions. Please check this and if necessary change dengue shock to dengue.

ABSTRACT: RESULTS: “Costs of VL impoverished approximately one-fifth of households”. Please rephrase

MINOR

“ABSTRACT AND AUTHOR SUMMARY: Introduction: Neglected tropical diseases (NTDs) mainly affect underprivileged populations, resulting in catastrophic health expenditure (CHE) and impoverishment from out-of-pocket (OOP) costs. This systematic review aimed to summarize the financial catastrophes from NTDs.” Not all NTDs lead to CHE. I would consider changing this to “potentially resulting in catastrophic health expenditure (CHE) and impoverishment from out-of-pocket (OOP) costs”

INTRO: “For example, as in the study reported before the first road map was published, every low-income country was affected by at least five NTDs.” – slightly unclear, please rephrase. 

In my opinion, the finding that “Financial catastrophes from NTDs were not comprehensively evaluated” should be a clear conclusion within the abstract/author summary (if this is possible with the word limit).

PLOS authors have the option to publish the peer review history of their article (what does this mean?). If published, this will include your full peer review and any attached files.

Reviewer #1: No

Reviewer #2: No

Figure Files:

Data Requirements:

Reproducibility:

References

---

## [Decision Letter · Decision Letter 1]

20 Mar 2024

Dear Prof. Chaiyakunapruk,

We are pleased to inform you that your manuscript 'Financial Hardship among Patients Suffering from Neglected Tropical Diseases: A Systematic Review and Meta-analysis of Global Literature' has been provisionally accepted for publication in PLOS Neglected Tropical Diseases.

Best regards,

Yoel Lubell

Guest Editor

Justin Remais

Section Editor

Reviewer's Responses to Questions

**Key Review Criteria Required for Acceptance?**

**Methods**

-Are the objectives of the study clearly articulated with a clear testable hypothesis stated?

-Is the study design appropriate to address the stated objectives?

-Is the population clearly described and appropriate for the hypothesis being tested?

-Is the sample size sufficient to ensure adequate power to address the hypothesis being tested?

-Were correct statistical analysis used to support conclusions?

-Are there concerns about ethical or regulatory requirements being met?

Reviewer #2: (No Response)

**Results**

-Does the analysis presented match the analysis plan?

-Are the results clearly and completely presented?

-Are the figures (Tables, Images) of sufficient quality for clarity?

Reviewer #2: (No Response)

**Conclusions**

-Are the conclusions supported by the data presented?

-Are the limitations of analysis clearly described?

-Do the authors discuss how these data can be helpful to advance our understanding of the topic under study?

-Is public health relevance addressed?

Reviewer #2: (No Response)

**Editorial and Data Presentation Modifications?**

Reviewer #2: (No Response)

**Summary and General Comments**

Reviewer #2: (No Response)

PLOS authors have the option to publish the peer review history of their article (what does this mean?). If published, this will include your full peer review and any attached files.

Reviewer #2: No

---

## [Editor Report · Acceptance letter]

22 Apr 2024

Dear Prof. Chaiyakunapruk,

We are delighted to inform you that your manuscript, "Financial Hardship among Patients Suffering from Neglected Tropical Diseases: A Systematic Review and Meta-analysis of Global Literature," has been formally accepted for publication in PLOS Neglected Tropical Diseases.

Best regards,

Shaden Kamhawi

co-Editor-in-Chief

Paul Brindley

co-Editor-in-Chief
